# CDTNet: Improved Image Classification Method Using Standard, Dilated and Transposed Convolutions

Yuepeng Zhou [1], Huiyou Chang [1,*], Yonghe Lu [2,*] and Xili Lu [3]

1   School of Computer Science and Engineering, Sun Yat-sen University, Guangzhou 510006, China; zhouyp9@mail2.sysu.edu.cn
2   School of Information Management, Sun Yat-sen University, Guangzhou 510006, China
3   School of Information and Engineering, Shaoguan University, Shaoguan 512005, China; luxili521@163.com
*   Correspondence: isschy@mail.sysu.edu.cn (H.C.); luyonghe@mail.sysu.edu.cn (Y.L.)

**Abstract:** Convolutional neural networks (CNNs) have achieved great success in image classification tasks. In the process of a convolutional operation, a larger input area can capture more context information. Stacking several convolutional layers can enlarge the receptive field, but this increases the parameters. Most CNN models use pooling layers to extract important features, but the pooling operations cause information loss. Transposed convolution can increase the spatial size of the feature maps to recover the lost low-resolution information. In this study, we used two branches with different dilated rates to obtain different size features. The dilated convolution can capture richer information, and the outputs from the two channels are concatenated together as input for the next block. The small size feature maps of the top blocks are transposed to increase the spatial size of the feature maps to recover low-resolution prediction maps. We evaluated the model on three image classification benchmark datasets (CIFAR-10, SVHN, and FMNIST) with four state-of-the-art models, namely, VGG16, VGG19, ResNeXt, and DenseNet. The experimental results show that CDTNet achieved lower loss, higher accuracy, and faster convergence speed in the training and test stages. The average test accuracy of CDTNet increased by 54.81% at most on SVHN with VGG19 and by 1.28% at least on FMNIST with VGG16, which proves that CDTNet has better performance and strong generalization abilities, as well as fewer parameters.

**Keywords:** CDTNet; dilated convolution; transposed convolution; feature fusion; receptive field

## 1. Introduction

Convolutional neural networks (CNNs) [1] have been widely applied in many fields, including image classification [2–6], natural language processing (NLP) [7], object detection [8–11], and speech classification [12]. Many CNN models have been developed and improved, and they have been successfully applied in medical fields [13,14], image denoising [15–17], and semantic segmentation [18–20].

The excellent performance of CNNs comes from their wider and deeper models [4]; however, these models have also faced an increasing memory burden [21], which limits their application in resource-constrained and high real-time requirement scenarios, such as mobile terminals and embedded systems with low hardware resources [22,23].

The CNN operation usually extracts features through the convolutional layer and integrates features by subsampling and fully connected (FC) layers; the method based on deep features can learn the most distinguishable semantic-level features from the original input [24]. Most image classification networks [2,3,25,26] employ successive pooling operations to gradually reduce the resolution of features and extend the receptive field (RF) size, but the pooling operations will cause information loss [27].

In CNNs, each feature map of the output only depends on a certain area of the input; a larger input area can capture more context information [15]. Enlarging the RF can extract

more contextual information [28]. The simple option is to stack successive convolutional layers or use a bigger size of filters, and the RF can be expanded, as mentioned in [29], but this often results in over-fitting because of large numbers of trainable parameters [13]. Some researchers [30–32] used various pruning methods to reduce the parameters, and these methods maintain or even improve the accuracy.

There are many other methods to alleviate the over-fitting problem of CNNs, such as L2-regularization [33], dropout [34], and data augmentation. Data augmentation is commonly adopted to alleviate the over-fitting problem [35] and reduce the need for regularization [36]. Data augmentation methods include translation [37], horizontal flipping [38], vertical flip and rotation, etc. Zheng et al. [39] used full stage data augmentation in their model and achieved a better performance.

Dilated convolution is a more effective method to expand the RF than the two methods mentioned above [15]. To reduce the computational complexity and improve the training speed, He et al. [40] used dilated convolution to expand the size of the RF without sacrificing the resolution. Some researchers [11,41] used dilated convolution for object detection. Heo et al. [42,43] used dilated convolution to effectively increase the receptive field in the source separation scheme, Lessmann et al. [13] used dilated convolution for automatic calcium scoring, and Xia et al. [14] used multi-scale dilated convolutions to extract richer features for computed tomography image segmentation.

Besides exploiting dilated convolution in the network, many works employed transposed convolution (TC) operations to realize high-resolution predictions to avoid information loss. TC layers were used to recover the spatial resolution [19,44]. Zeiler et al. [45] used TC to recover low-resolution prediction maps. Qu et al. [41] used TC to enrich low-level features and achieved superior performance in terms of the high detection rate. Shelhamer et al. [18] introduced TC for semantic segmentation.

We present an architecture to fuse different features of standard convolution, dilated convolution, and TC (CDTNet) for image classification, which takes the VGG model [3] as the framework. The general idea of CDTNet is to capture richer information without increasing the number of parameters. We combined the standard and dilated convolution to extract multi-scale features and used transposed convolution to transmit features from low level to high level, which can recover part of the lost information in the pooling layers. Through the combination of these methods, the number of parameters can be decreased while reducing the loss value and improving the accuracy. To the best of our knowledge, we are the first to apply standard, dilated, and transposed convolutions together for image classification. We used three image datasets, CIFAR-10, SVHN, and FMNIST, to evaluate all models for convergence speed, loss, and accuracy.

There are many popular image classification models, such as AlexNet, VGG, GoogLeNet, YOLO, ResNet, ResNeXt, DenseNet, and so on. We used VGGs (VGG16 and VGG19), ResNeXt [46], and DenseNet [8] as our baseline models, with successive $3 \times 3$ convolution operations to build networks. The main contributions of this article can be summarized by the following three aspects.

We propose a more powerful model for image classification by assembling standard, dilated, and transposed convolutions, which can considerably improve the performance. The CDTNet can better represent images by integrating low-level and high-level features.

The dilated convolution can increase the RFs, and the TC can increase the spatial size of the feature maps to recover low-resolution prediction maps. The feature maps of different levels are integrated to obtain multi-scale context information, which improves the classification ability of the network.

The CDTNet exhibits robustness and rapid convergence speed. All evaluation metrics of CDTNet are better than the baseline models, namely, VGGs, ResNeXt, and DenseNet.

The remainder of this paper is organized as follows. In Section 2, we provide a brief overview of the relevant literature. Section 3 introduces details of CDTNet. In Section 4, we describe the experimental settings, datasets, and present the results from our models and compared models. In Section 5, we present our conclusions.

## 2. Related Works

### 2.1. Dilated Convolution

Dilated convolution was first used in [43], called atrous convolution. Yu et al. [47] called the same operation dilated convolution in their article; dilated convolution has a larger RF than the standard convolution, but the number of weight parameters is the same [48], which is shown in Figure 1 [49].

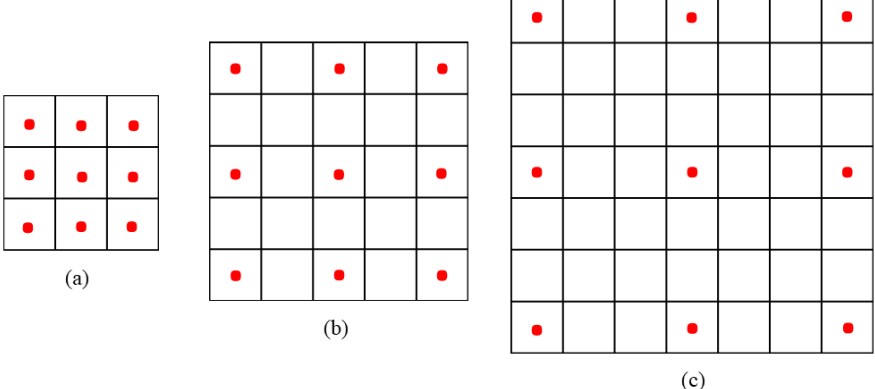

**Figure 1.** The RF of different dilation rates at 3 × 3 filter. (**a**) Dilation rate = 1, the RF is 3 × 3. (**b**) Dilation rate = 2, the RF is 5 × 5. (**c**) Dilation rate = 3, the RF is 7 × 7.

From Figure 1, we can see that we can control the RF of the models conveniently to use dilated convolution. When the dilation rate is 1, dilated convolution is standard convolution. When the dilation rate is 2 or 3, the size of the RF is 5 × 5 or 7 × 7, respectively, as presented in Figure 1a–c. All the values are zero except the value at the red dot.

Dilated convolution with rate $r$ introduces $r - 1$ zeros between successive filter parameters, which can enlarge the kernel size of the filter effectively. The RF can be calculated by the following formula:

$$rf = k + (k - 1) * (r - 1) \tag{1}$$

where $k$ represents the filter size, and $r$ is the dilated rate.

Compared with purely convolution networks, dilated convolution can capture richer information. The number of parameters in dilated convolution is the same as that in the standard convolution, but the size of the RF increases linearly [14].

Dilated convolution layers with different dilated rates can fetch multi-scale features. To capture more contextual information at multiple scales, Lu et al. [49] and Xia et al. [14] set the dilation rates as 1, 3, and 5 to extract richer features. DeepLabv2 [20] (rate = 6, 12, 18, 24) and Yao et al. [50] proposed PYolo to use multi-branch convolution with dilation rates of 1, 3, 6, and 12 for detecting pneumonia; these features complement each other to ensure that the information distributed in different ranges can be sampled [51].

### 2.2. Transposed Convolution

TC was introduced by Zeiler et al. [52] and is widely used in generative models for computer vision [53,54]. TCs work by exchanging the backward and forward passes of a convolution [55]. TC is used to increase the spatial size of the feature maps to recover the low-resolution prediction maps. The process of standard and transposed convolutions is shown in Figure 2 [56].

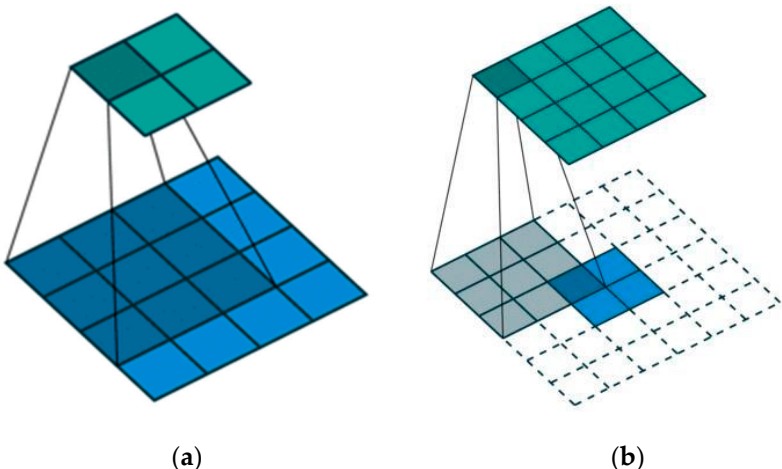

**Figure 2.** Standard convolution (**a**) and TC (**b**) (blue is input, shadow is filter size, green is output).

TC is not exactly the reverse operation of standard convolution, but it reconstructs the high-dimensional state by gradually up-sampling low-dimensional representations [57].

By constructing a transposed model, the low-resolution features can be mapped to the high-resolution ones, and accurate boundary location can be generated through pixel-level supervision [58].

For example, refs. [18,59–61] employed TC after the pooling layer to enlarge the size of the low-resolution features and make the size of the output the same as the input. Zeiler et al., used TC in computer vision to capture mid- and high-level image structures [45], and Gulrajani and Yu et al., used TC to generate high-resolution feature maps [53,54], achieving remarkable performance in the up-sampling process.

### 2.3. Feature Fusion Methods

Feature fusion is an important operation in the CNN models, which can transmit the information of lower layers directly to higher layers [21]. Merging the features from different layers can achieve the effect of aggregating information from different RFs. For the operation method of the joining layer, many researchers chose addition and concatenation, both of which seem reasonable [9].

The experimental results of [61] showed that the concatenation operation is more effective in their architectures. Fu et al. [58] adopted concatenation on the outputs of two branches, which brings too much memory demand, and they performed a convolution with fewer filters to reduce the filter number of the feature map. Li et al. [29] used two dense dilated blocks with dilated convolution; the features extracted from the last three blocks were concatenated as the input of the inference layer for multi-scale attention competition. Many other researchers have utilized concatenation to fuse features [16,62–64].

However, the concatenation operation naturally raises the processing time. Since the input channels of each layer remain unchanged, the running time of the models using element-wise summation in the skip connection layer is almost equal to that of the model without skip connection [51]. Xie et al. [46] used summation operation to merge features, and many other researchers have also used summation to fuse features [24,51,65–67].

Unlike Huang and Xie et al., Dai [68] used concatenation operation to combine multi-scale features, then used sum operation after blocks to supplement the missing information in the pooling layers.

### 2.4. Skip Connection

In CNN, deeper layers can capture global features by stacking convolutional layers, but these cannot prove that the features extracted by the last layer are the final representation for any task [69]. This indicates that combining information of low and high layers can

yield the contextual and abstraction information of objects, which can improve the accuracy of image super-resolution restoration.

Different intermediate layers can extract different semantic levels and RFs; the merged features through the skip connections contain contextual and abstraction features that are extracted in different blocks [51]. Skip connection is a suitable method to combine the local and global features to strengthen feature propagation [9,61], which can describe different sizes objects comprehensively. The skip connection can also avoid the gradient vanishing, which can benefit back propagation [17] and provide rich information flow to the next layer.

Ronneberger et al., added skip connections between the encoder and the corresponding size decoder in their proposed model, U-Net [44]. Yu et al. [69] showed that the skip connection method is an effective method to make the following layers acquire the information from the previous layers. Shelhamer et al. [18] used skip connections to connect the coarse granularity features with the fine granularity features to improve the prediction effect.

## 3. Fuse Different Features of CDTNet

Inspired by the previously mentioned research, we proposed CDTNet to fuse different features of standard, dilated, and transposed convolutions for image classification. Similar to the VGG models, we used $3 \times 3$ filters and doubled the filter numbers after every pooling operation [3], except the last one.

In our model, we used two parallel convolution operations in the beginning stage: one uses standard convolution and the other uses dilated convolution. The results of the two branches are fused by a concatenation operation, which naturally raises the dimension. After the concatenating operation, we used a block containing $1 \times 1$ convolution, Batch Normalization (BN) [70], ReLU [71], and $2 \times 2$ max-pooling (CBRP), four consecutive operations, to reduce the channel number of the feature map. The process of standard and dilated convolutions is shown in Figure 3.

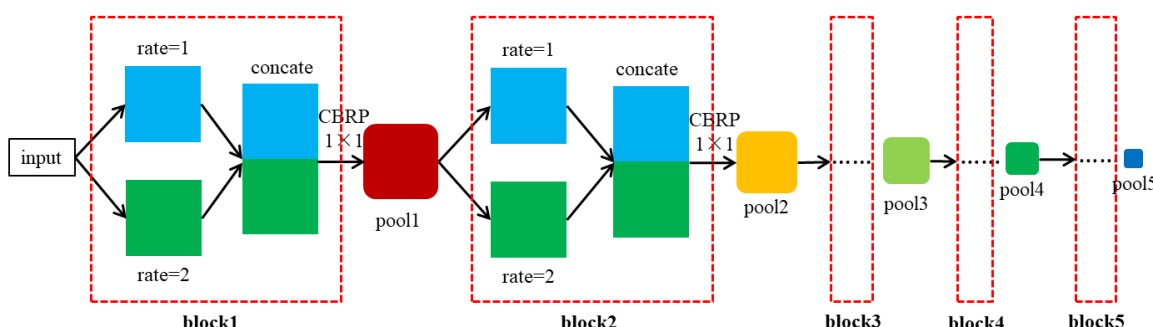

**Figure 3.** Process of standard and dilated convolution.

The CDTNet comprises five blocks in Figure 3, and each block contains three parts: standard and dilated convolutions, concatenate, and CBRP operations. The usage of standard and dilated convolution layers can capture image clues of multiple scales by expanding the RF and can avoid increasing the number of model parameters. In each block, rate = 1 represents standard convolution, and rate = 2 represents that the dilation rate is 2 in the dilated convolution operation. The concatenation operation can combine the features from two branches, which represent the features of the global and local images. After each concatenation operation, we added a CBRP operation to extract features. The size of the convolutional kernel is $1 \times 1$, which has been used to reduce the parameters of the network and computational costs. In addition, the third dimension of feature maps, i.e., the number of channels, is controlled by the number of the $1 \times 1$ filter.

It is useful to increase the dilation rate moderately for better performance [61]. Xia et al. [14] used four dilated rates to extract features, then fused the four levels of features to make full use of the low-level and high-level features. Larger RF can be obtained by enlarging the dilation rate; however, as the filling size increases with the dilation rate, the

boundary effect is introduced, which counterbalances the effect of large RF obtained by increasing the dilation rate [61]. We used two dilation rates in CDTNet, i.e., rate = 1 and rate = 2. There are five blocks with the process of standard and dilated convolutions, and the output size of each block is half the size of the front block's output.

The output of each block is fed into up-sampling units for finer information recovery, which can generate high-resolution features. The compared experiments of [61] suggest that the features extracted with a small upscaling factor could retain more detailed information. Thus, we used the filter number in the block: divide by 2 for ×2 TC, divide by 4 for ×4 TC, and so on. The last two TCs have the same filter number. All TC results of the same size are concatenated together with the same size pooling result; the process is shown in Figure 4.

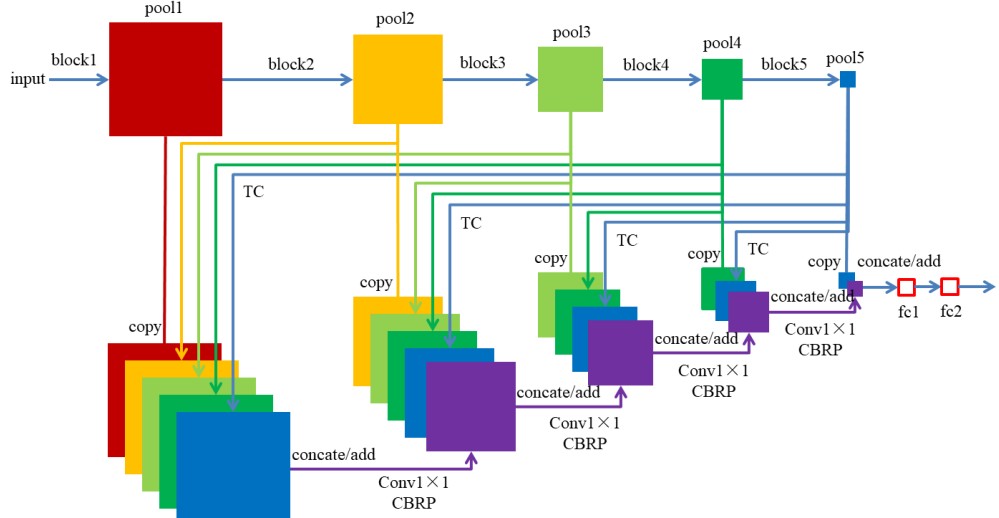

**Figure 4.** CDTNet architecture.

In Figure 4, pool1 to pool5 correspond to pool1 to pool5 and block1 to block5 correspond to block1 to block5 in Figure 3. The polylines with an arrow represent TC operations, which are marked with "TC", and the straight lines under each pool block without arrows represent skip connections, which are marked with "copy". The pooling operation will abandon some important feature information. The skip connection is widely used in many popular deep networks, and the advantage is that it allows more lower-level information to reach the top level. We used skip connection in CDTNet, and the fused feature map retains the high resolution of the lower-level feature map and represents better semantic information.

In Figure 4, the feature maps in previous layers are fused with other TC results. We used two methods to fuse the features, concatenation (CDT_C) and addition (CDT_A). The details of the fuse process in the lower left part of Figure 4 are shown in Figure 5.

In Figure 5, each pooling result is transposed with different strides and filter numbers, where TC represents transposed convolution, s is stride, and num is filter number. Several successive max-pooling and transpose operations may lead to the information loss of low-level features, so we set the filter number by dividing by 2 in order, except for the last TC process, where the filter number equals that of the second-to-last TC.

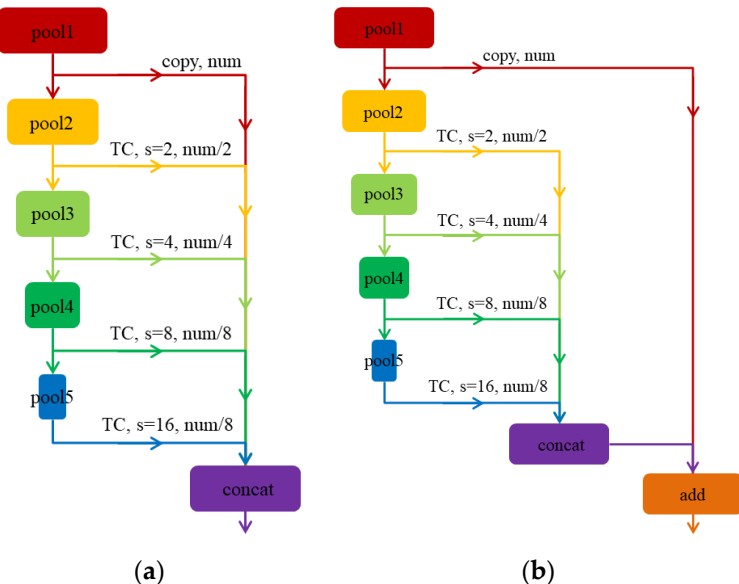

<p style="text-align:center;">(<b>a</b>)      (<b>b</b>)</p>

**Figure 5.** Feature fused methods. (**a**) Concatenate the features of TC results and skip connection. (**b**) Sum the features of TC results, then concatenate with skip connection.

After TC, the outputs of pool2 to pool5 have the same size as pool1. The feature maps of all transposed results and pool1, in Figure 5a, are concatenated into a single tensor as the input of the next layer.

$$O_i = \left[ p_i, p_{i+1} t^{2^1}, \; p_{i+2} t^{2^2} \ldots \ldots, \; p_{i+n} t^{2^n} \right] \tag{2}$$

where $O_i$ concatenates all feature maps, and $p_{i+n} t^{2^n}$ is the result of $(n)$-th pooling layer transposed with stride $2^n$.

The feature maps of all transposed results in Figure 5b are concatenated into a single tensor, and then added with pool1 as input of the next layer.

$$C_i = \left[ p_{i+1} t^{2^1}, \; p_{i+2} t^{2^2} \ldots \ldots, \; p_{i+n} t^{2^n} \right] \tag{3}$$

$$O_i = \frac{add(p_i, C_i)}{2} \tag{4}$$

where $C_i$ concatenates all transposed feature maps except $p_i$, and $O_i$ averages $p_i$ and $C_i$.

According to the output size, each pooling result performs different times of TC. The TCs of pooling results are shown in Figure 6.

In Figure 6, pool5 is transposed four times, and the stride is 2, 4, 8, and 16, respectively. The number of channels is 512, 128, 32, and 8, which is based on the filter number of each convolutional layer and the ratio in Figure 5.

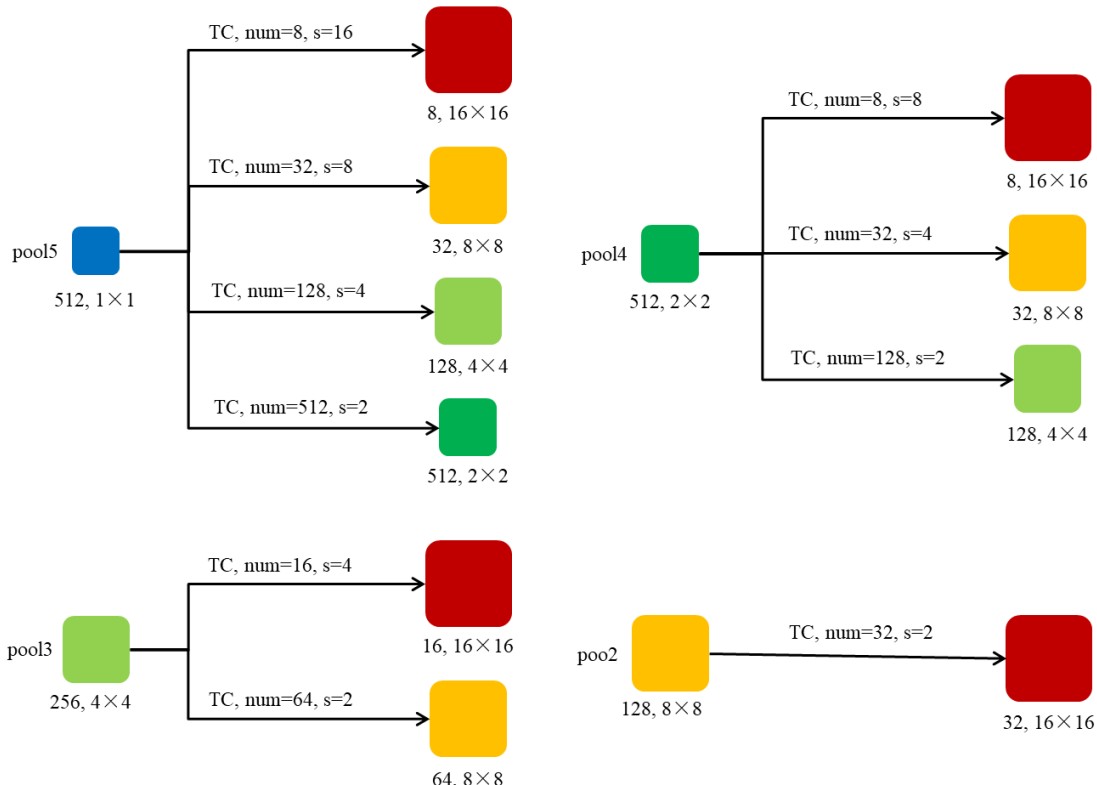

**Figure 6.** TC of each pooling result.

## 4. Experiments

We compared CDTNet with four classical classification models—VGG16, VGG19, ResNeXt, and DenseNet—to evaluate the proposed approach objectively and comprehensively. Extensive experiments were carried out on three challenging benchmarks, i.e., CIFAR-10 [72], SVHN [73], and FMNIST [74].

All experiments were conducted on a server with Intel Xeon Gold 6139 M (2.3–3.7 GHz) processors, 88 GB memory, and an NVIDIA GeForce RTX 2080 Ti graphics card. The operating system was 64 Bit Ubuntu 16.04. Tensorflow was used for building the model, and the main source codes of ResNeXt and DenseNet were taken from the websites (https://github.com/taki0112/ResNeXt-Tensorflow, accessed on 12 July 2021) and (https://github.com/taki0112/Densenet-Tensorflow, accessed on 12 July 2021), respectively.

### 4.1. Datasets

CIFAR-10: The CIFAR-10 [72] dataset contains 60,000 color images from 10 different classes: trucks, cats, cars, horses, airplanes, dogs, ships, deer, birds, and frogs. The size of these images is 32 × 32 pixels. The dataset contains 50,000 images for training (5000 images in each category) and 10,000 images for testing.

SVHN: The Street View House Number (SVHN) [73] dataset comprises color images of house numbers, collected by Google Street View. SVHN comprises 73,257 training images and 26,032 test images. The digits 0 to 9 offer a multi-class classification with resolution of 32 × 32 pixels. The SVHN shows vast intra-class variations and includes complex photometric distortions, which makes the recognition problem a challenge [75].

Fashion-MNIST: The Fashion-MNIST [74] dataset comprises 28 × 28 grayscale images, with 70,000 fashion products belonging to 10 categories: T-shirts/tops, pullovers, trousers, dresses, coats, sandals, sneakers, shirts, bags, and ankle boots. The dataset contains 60,000 training images and 10,000 test images.

*4.2. Parameter Settings*

Parallel convolutional layer: $3 \times 3$ convolutional kernel has been proven to be the most effective kernel size for natural images [76]. We used $3 \times 3$ convolutional kernels and 1-padding with stride 1 to guarantee that the size of the outputs equals that of the inputs. In the dilated convolution channel, the dilated rate is 2. The outputs of the two channels are fused by the concatenation method. BN leads to considerable improvements in convergence while eliminating the need for other forms of regularization; every convolution operation is followed by a BN operation.

Pooling layer: We used max-pooling with a $2 \times 2$ pixel window, and stride is 2 in each pooling operation.

TC layer: For CIFAR-10 and SVHN, the input size is $32 \times 32$, and the output size is divided by 2 after each max-pool. In the TC layer, the input size is multiplied by 2, 4, 8, and 16 for pool5 to recover the size corresponding to pooling layers, and by 2, 4, and 8 for pool4, and so on.

Feature fusion layer: Based on the output size of TC and pooling layers, we used the concatenation (CDT_C) and addition (CDT_A) method to fuse the features.

FC layer: Because the FC layers in VGGs have a large number of redundant parameters, according to the ablation experiments of [33], we added two FC layers in our model; the neuron numbers are 1024 and 10.

The last FC layer is connected with a 10-class layer with cross-entropy loss. Softmax was selected to obtain the category probability, formulated as Formula (5):

$$p(y|x) = \frac{exp\left(w_y \cdot x + b\right)}{\sum_{c=1}^{C} exp(w_c \cdot x + b)} \tag{5}$$

where $C$ is the number of channels, $w \in R^{C \times N}$, $N$ is the number of classes, and $p(y|x) \in R^N$ is the scaled classification score.

On the CIFAR-10 dataset, we used Nesterov momentum with a momentum weight of 0.9, and a weight decay of 0.0003. All models were trained with an initial learning rate of 0.1, divided by a factor of 10 after 80 and 120 epochs, and the batch size was 250. On the SVHN dataset, the models were trained for 50 epochs, and the batch size was 96. On the FMNIST dataset, all models were trained for 50 epochs with a batch size of 100. We used the Adam optimization method and set the learning rate as 0.001 for SVHN and FMNIST datasets.

We adopted data augmentation methods such as translation [37] and horizontal flipping [38] for CIFAR-10 and SVHN datasets, and used L2-regularization [34] techniques for limiting network complexity.

*4.3. Results and Discussion*

We used four evaluation metrics (training loss, training accuracy, test loss, test accuracy) for CIFAR-10 and SVHN, and three evaluation metrics (training loss, training accuracy, test accuracy) for FMNIST. We compared the experimental results of CDTNet with VGGs, ResNeXt, and DenseNet. At the same time, we also compared the parameters of these models. The parameters can be calculated as Formula (6) [21]:

$$P = fn_f * C^2 * fn_n \tag{6}$$

where $P$ represents the parameters; $fn_f$ and $fn_n$ denote the filter number of the front and next layers, respectively; and $C$ represents the filter size.

4.3.1. CIFAR-10

We performed the experiments using six models (VGG16, VGG19, ResNeXt, DenseNet, CDT_C, and CDT_A) on CIFAR-10. Four evaluation metrics were used to evaluate the performances of the models. The experimental comparison results are plotted in Figure 7.

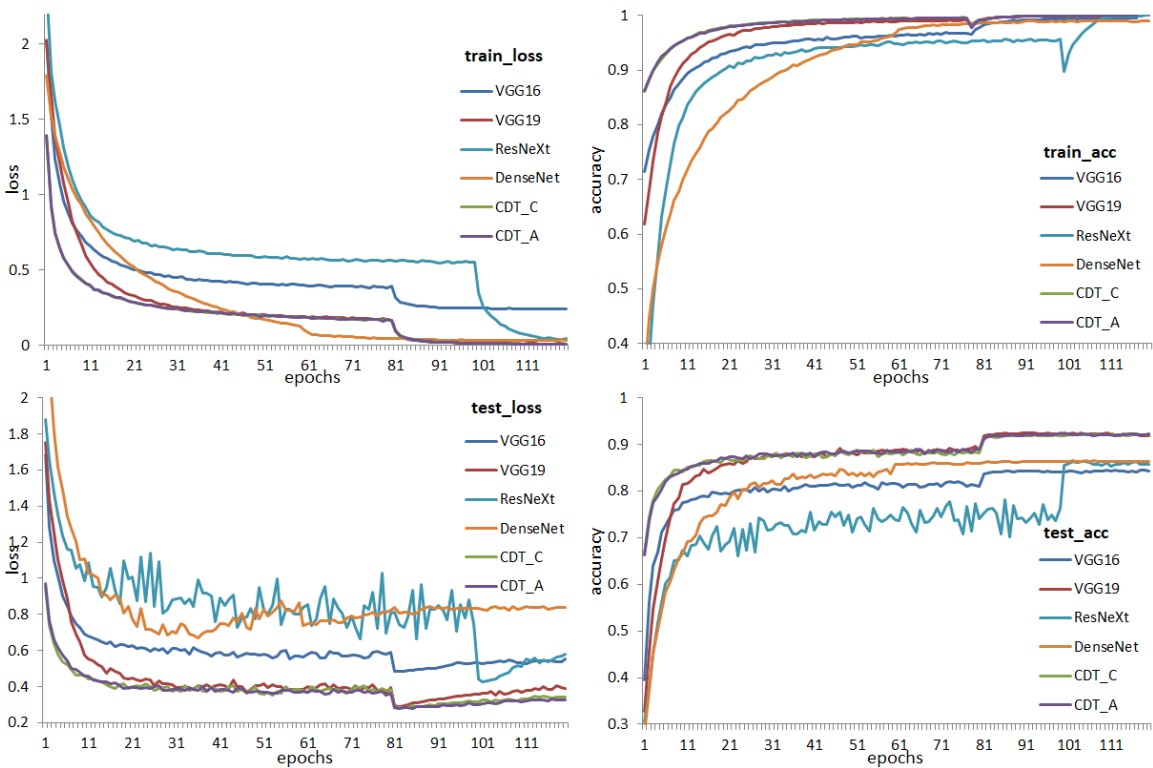

**Figure 7.** Compared results on CIFAR-10.

Figure 7 reveals that CDTNet has a better performance in the training and test stages. CDT_C improves the average test accuracy by 9.43%, 1.48%, 19.92%, and 8.89% compared to VGG16, VGG19, ResNeXt, and DenseNet, respectively. CDT_A improves the average test accuracy by 9.61%, 1.65%, 20.13%, and 9.07% compared to VGG16, VGG19, ResNeXt, and DenseNet, respectively. They also reduce the average training loss and improve the average training accuracy. The specific values are shown in Table 1, where the symbol ↓ indicates a reduction and the symbol ↑ indicates an improvement.

**Table 1.** Specific compared results on CIFAR-10.

| Model | Baseline | Training Loss | Training Accuracy | Test Loss | Test Accuracy |
|-------|----------|---------------|-------------------|-----------|---------------|
| CDT_C | VGG16 | 55.3% ↓ | 3.99% ↑ | 36.76% ↓ | 9.43% ↑ |
| CDT_C | VGG19 | 20.3% ↓ | 2.42% ↑ | 14.1% ↓ | 1.48% ↑ |
| CDT_C | ResNeXt | 67.17% ↓ | 6.84% ↑ | 54.36% ↓ | 19.92% ↑ |
| CDT_C | DenseNet | 28.4% ↓ | 7.61% ↑ | 56% ↓ | 8.89% ↑ |
| CDT_A | VGG16 | 55.15% ↓ | 3.96% ↑ | 37.57% ↓ | 9.61% ↑ |
| CDT_A | VGG19 | 20.05% ↓ | 2.39% ↑ | 15.19% ↓ | 1.65% ↑ |
| CDT_A | ResNeXt | 67.06% ↓ | 6.81% ↑ | 54.94% ↓ | 20.13% ↑ |
| CDT_A | DenseNet | 28.16% ↓ | 7.58% ↑ | 56.6% ↓ | 9.07% ↑ |

### 4.3.2. SVHN

We performed the same experiments on SVHN as those on CIFAR-10, but the performance of VGGs was very poor, so we modified the kernel size to 1 for layers 7, 10, and 13 in VGG16, and for layers 8, 12, and 16 in VGG19. Figure 8 shows the compared experimental results on the SVHN dataset.

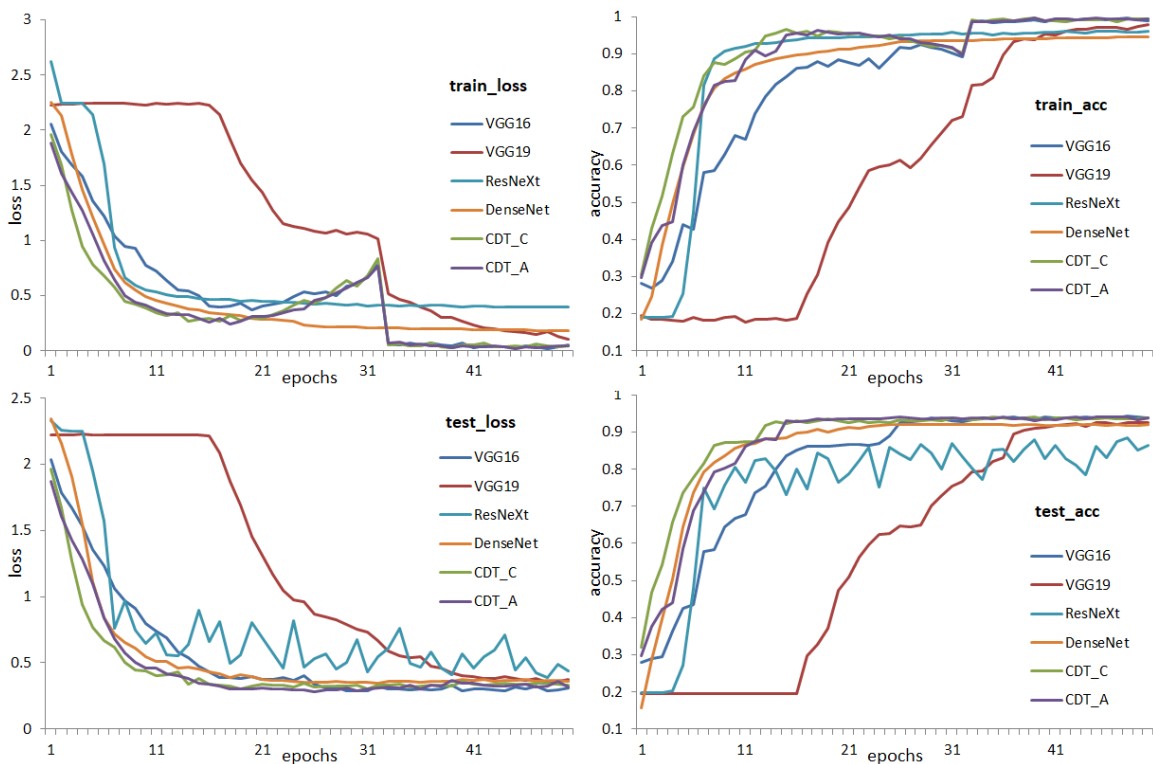

**Figure 8.** Compared results on SVHN.

From Figure 8, we can draw the same conclusion that the performance of CDTNet is better than that of VGGs, ResNeXt, and DenseNet. CDT_C improves the average test accuracy by 9.13%, 54.81%, 17.49%, and 3.82% compared to VGG16, VGG19, ResNeXt, and DenseNet, respectively. CDT_A improves the average test accuracy by 6.79%, 51.48%, 14.97%, and 1.59% compared to VGG16, VGG19, ResNeXt, and DenseNet, respectively. They also reduce the average training loss and improve the average training accuracy. The specific values are shown in Table 2.

**Table 2.** Specific compared results on SVHN.

| Model | Baseline | Training Loss | Training Accuracy | Test Loss | Test Accuracy |
|-------|----------|---------------|-------------------|-----------|---------------|
| CDT_C | VGG16 | 25.55% ↓ | 9.83% ↑ | 18.14% ↓ | 9.13% ↑ |
| CDT_C | VGG19 | 68.83% ↓ | 58.73% ↑ | 62.85% ↓ | 54.81% ↑ |
| CDT_C | ResNeXt | 42.16% ↓ | 5.54% ↑ | 40.8% ↓ | 17.49% ↑ |
| CDT_C | DenseNet | 13.84% ↓ | 5.92% ↑ | 17.68% ↓ | 3.82% ↑ |
| CDT_A | VGG16 | 23.3% ↓ | 7.71% ↑ | 15.98% ↓ | 6.79% ↑ |
| CDT_A | VGG19 | 67.89% ↓ | 55.67% ↑ | 61.87% ↓ | 51.48% ↑ |
| CDT_A | ResNeXt | 40.41% ↓ | 3.50% ↑ | 39.24% ↓ | 14.97% ↑ |
| CDT_A | DenseNet | 11.24% ↓ | 3.87% ↑ | 15.51% ↓ | 1.59% ↑ |

### 4.3.3. FMNIST

Because the size of FMNIST is 28 × 28, which is not the same as CIFAR-10 and SVHN, when we transposed the features from blocks 3 and 4, the feature size of TC layers was not equal to that of previous layers, so we used additional layers with stride 2 and 0-padding to adjust the output size to equal that of previous features that will lose some boundary information. The experimental results are shown in Figure 9.

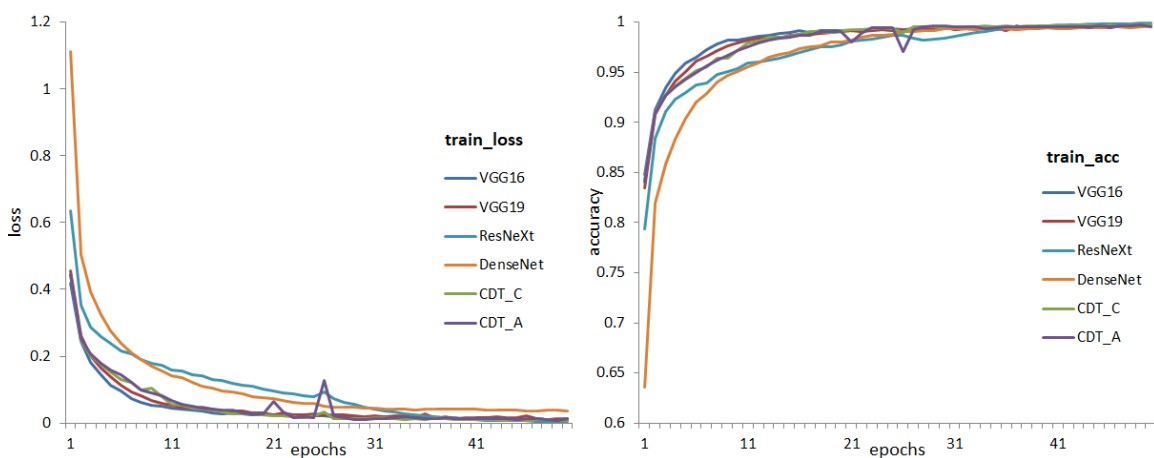

**Figure 9.** Compared results on FMNIST.

As seen in Figure 9, the additional 0-padding convolution may lose some boundary information, and the training loss and training accuracy of VGGs are better than those of CDTNet at the beginning, but the performance of CDTNet turns the tide after epoch 28, and the test accuracy of CDTNet is also better than VGG16, VGG19, ResNeXt, and DenseNet. The test accuracies of all models are shown in Table 3.

**Table 3.** Accuracies of CDTNet and other models on FMNIST.

| Models | VGG16 | VGG19 | ResNeXt | DenseNet | CDT_C | CDT_A |
|---|---|---|---|---|---|---|
| Accuracy | 0.9213 | 0.9207 | 0.9172 | 0.9133 | 0.9337 | 0.9331 |

The CDTNet has better performance in the training stage and improves the test accuracy. CDT_C improves the test accuracy by 1.35%, 1.41%, 1.80%, and 2.23% compared to VGG16, VGG19, ResNeXt, and DenseNet, respectively. CDT_A improves the test accuracy by 1.28%, 1.35%, 1.73%, and 2.17% compared to VGG16, VGG19, ResNeXt, and DenseNet, respectively.

### 4.3.4. Parameter of Models

The parameters of these models are shown in Table 4 according to Figures 3–6 and Formula (6).

**Table 4.** Parameters of CDTNet and classical models.

| Model | VGG16 | VGG19 | ResNeXt [46] | DenseNet [9] | CDT_C | CDT_A |
|---|---|---|---|---|---|---|
| Parameters (M) | 32.06 | 37.13 | 23.84 | 27.2 | 24.18 | 16.79 |

From Table 4, we can see that there are slightly more parameters for CDT_C than ResNeXt, but the number of parameters of CDT_A is much less than other models.

To sum up, through the experimental results of the above three datasets, it can be seen that the CDTNet reduces the training and test losses and improves the accuracies. There are outliers during the training stage on FMNIST because we used additional layers to adjust the feature size, resulting in some boundary information lost, but this does not affect the overall performance of CDTNet. The average test accuracy of CDTNet increased by 54.81% at most on SVHN with VGG19 and by 1.28% at least on FMNIST with VGG16.

## 5. Conclusions

In this study, we proposed CDTNet with standard, dilated, and transposed convolutions. The standard and dilated convolution can extract multi-scale features, and the



transposed convolution can transmit features from low level to high level, which can recover part of the lost information in the pooling layers. Because the object size is small, we used a dilated rate of 2 to fetch the features to concatenate the output of standard convolution. Each block except block 1 was followed by a transposed operation to increase the spatial size of the feature maps to recover low-resolution prediction maps.

We evaluated the model on CIFAR-10, SVHN, and FMNIST datasets with VGG16, VGG19, ResNeXt, and DenseNet. CDTNet improves the average test accuracy by 1.48% to 20.13% and reduces average test loss by 14.1% to 56.6% on CIFAR-10. On SVHN, CDTNet improves the average test accuracy by 1.59% to 54.81% and reduces the average test loss by 15.51% to 62.85%. On FMNIST, CDTNet improves the average test accuracy by 1.28% to 2.23%. The experimental results show that all evaluation metrics of CDTNet are better than those of the state-of-the-art models, which proves that CDTNet has better performance and strong generalization abilities—and fewer parameters.

In future work, we will explore more effective architecture to fuse different granularity features and adopt diversified evaluation metrics to analyze the performance. In addition, as not all input image sizes are to the nth power of 2, in future work, we will explore a more effective method to set the number of TC channels and design the feature size after TC operation.

**Author Contributions:** Conceptualization, H.C. and Y.L.; methodology, Y.Z.; software, X.L.; validation, Y.Z., H.C., and Y.L.; formal analysis, H.C.; writing—original draft preparation, Y.Z.; writing—review and editing, X.L.; visualization, X.L.; supervision, H.C.; project administration, Y.L.; funding acquisition, H.C. and Y.L. All authors have read and agreed to the published version of the manuscript.

**Funding:** This research was funded by the Basic and Applied Basic Research Fund of Guangdong Province, grant number 2019B1515120085.

**Informed Consent Statement:** Not applicable.

**Conflicts of Interest:** The authors declare no conflict of interest.

## Abbreviations

CBRP: convolutional, batch normalization, ReLU, pooling; CDT: convolution, dilated, transposed; CNN: convolutional neural network; DenseNet: Dense Convolutional Network; FC: fully connected; MNIST: Modified National Institute of Standards and Technology; FMNIST: Fashion-MNIST; NLP: natural language processing; NSFC: Natural Science Foundation of China; RF: receptive field; ResNet: Residual Neural Network; ResNeXt: the Next Dimension of ResNet; SVHN: Street View House Number; TC: transposed convolution; VGG: Visual Geometry Group; YOLO: You Only Look Once.

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
