# Peer review of "CDTNet: Improved Image Classification Method Using Standard, Dilated and Transposed Convolutions"

_applsci, doi:10.3390/app12125984_

Round 1
Reviewer 1 Report
Dear Authors
I appreciate the opportunity to contribute with your interesting work. I will mention some opportunities for improvement in each section.
Introduction
The Introduction is correctly stated . But it would be good to establish a general objective for the work.
Theoretical Framework
The theoretical framework is well updated
The proposed Method is clear but it is necessary It is necessary to review formulas (2), (3) and (4) and include the references of the formulas and verify that the equations are correct.
The results are good and consistent with the proposed method.
Discussion and conclusions
It would be good to include a discussion of the results obtained in the investigation and to compare those results with other similar investigations.
References
It is better to provide an up-to-date literature review. There are excellent very current examples published on the subject of Applied Sciences Journal or in other MDPI journals.
Author Response
Thanks for you comments. Please see the attachment.
Reviewer 2 Report
The paper presents the fusion of different features of standard convolution for image classification and uses different benchmark datasets to analyze convergence speed and accuracy.
The abstract should be modified since it does not summarize the objectives of the work presented. The article is long and somewhat difficult to follow, although the authors have taken care to explain the whole process in detail. The bibliographic review made by the authors is correct and very exhaustive.
The idea of performing the fusion process seems interesting and is presented in a reasonably easy-to-understand graphical form. The benchmarks used seem to me to be adequate for verification and the results presented show in some cases considerable improvements and in others more modest values. In my opinion, the article could be accepted with a minor revision introducing these modifications:
1. as I stated above the abstract needs to be reworded in a way that summarizes the objectives and results of the paper.
2. The conclusion section is also poor. It should be more detailed.
3. Although the authors are thorough to the point of indicating the hardware they have used, at least as far as I am able to understand, they do not indicate whether the fusion process increases computational time. It would be highly desirable that this aspect be made clear and included as part of the conclusions.
4. There is a small detail regarding the format. There is no space between the figure captions and the next paragraph.
Author Response

(The authors gave the same response as above.)

Reviewer 3 Report
1- What about running time (execution time) of the methods?
Additional results (a new table or chart/graph) may also be given in terms of running times.
2- A concern is that no formal statistical analysis of the results are done, to indicate whether the differences in performance are statistically significant or not.
For example; Friedman Aligned Rank Test, Wilcoxon Test, Quade Test, etc.
p-value can be calculated and compared with the significance level (p-value < 0.05).
3- This paper is about "image classification". YOLO is also a useful and popular method that has been successfully used for image classification. Hovewer, this paper does not mention about YOLO. I suggest the authors citing the following recent study related to YOLO ("Shelf Auditing Based on Image Classification Using Semi-Supervised Deep Learning to Increase On-Shelf Availability in Grocery Stores").
4- The authors may explain the possible future studies in the conclusion section.
5- Some abbreviations are used in the text without giving their expansion.
For example; ResNet, ResNeXt, etc.
The authors should write that "these abbreviations stand for what".
6- The symbols in the text should be italic.
For example:
- "r introduces r-1 zeros between successive filter values"
- "where k and r denote the filter size"
- "where C is the number of channels"
- "N is the number of classes"
- ... etc.
Author Response

(The authors gave the same response as above.)

Round 2
Reviewer 3 Report
The authors revised the manuscript adequately according to the reviewers comments.
The manuscript is now more qualified and clear.
I have no further comments.
I suggest accepting it for publication as it stands.